# High-throughput small molecule screen identifies inhibitors of microsporidia invasion and proliferation in *C. elegans*

Brandon M. Murareanu[1], Noelle V. Antao[2], Winnie Zhao [1], Aurore Dubuffet[3], Hicham El Alaoui[3], Jessica Knox [1,4], Damian C. Ekiert [2,5], Gira Bhabha[2], Peter J. Roy [1,4] & Aaron W. Reinke [1] ✉

Microsporidia are a diverse group of fungal-related obligate intracellular parasites that infect most animal phyla. Despite the emerging threat that microsporidia represent to humans and agricultural animals, few reliable treatment options exist. Here, we develop a high-throughput screening method for the identification of chemical inhibitors of microsporidia infection, using liquid cultures of *Caenorhabditis elegans* infected with the microsporidia species *Nematocida parisii*. We screen a collection of 2560 FDA-approved compounds and natural products, and identify 11 candidate microsporidia inhibitors. Five compounds prevent microsporidia infection by inhibiting spore firing, whereas one compound, dexrazoxane, slows infection progression. The compounds have in vitro activity against several other microsporidia species, including those known to infect humans. Together, our results highlight the effectiveness of *C. elegans* as a model host for drug discovery against intracellular pathogens, and provide a scalable high-throughput system for the identification and characterization of microsporidia inhibitors.

Microsporidia are a diverse group of parasites, comprising over 1400 species that can infect most major animal phyla[1–3]. Many microsporidia species infect agriculturally important animals. This includes invertebrate-infecting species such as *Nosema ceranae* and *Nosema apis* which infect honey bees, *Enterocytozoon hepatopenaei* which infects shrimp, *Hepatospora eriocheir* which infects crabs, and *Nosema bombycis* which infects silkworms[4–7]. Additionally, there are several species that infect farmed fish, including *Loma salmonae* which infects salmon and trout, as well as *Enterospora nucleophila* which infects gilthead sea bream[8,9]. Microsporidia infection in these animals can result in reduced body size, fewer offspring, and increased mortality[4–9]. The economic impact of microsporidia infections is high. The infection of silkworms has triggered historical collapses of the sericulture industry, and microsporidia infections are estimated to

cost over $200 million USD annually to Thailand's shrimp industry[7,10]. Livestock such as pigs, cattle, and sheep, as well as pets such as dogs, cats and rabbits are infected by *Encephalitozoon* species and *Enterocytozoon bieneusi*. Humans are also infected by these species, with infections in the immunocompromised being more prevalent[11]. Microsporidia commonly infect animals, with over half of honeybee hives and approximately 40% of pigs reported to be infected[11,12]. Microsporidia are also emerging pathogens, with the threat posed by many of these species only being recognized in the last several decades[13].

Despite the threat that these parasites pose to human and animal health, few therapeutic options exist. Fumagillin is a compound from the fungus *Aspergillus fumigatus* which inhibits methionine aminopeptidase type 2 (MetAP2), and has been used since the 1950s to treat

[1]Department of Molecular Genetics, University of Toronto, Toronto, ON, Canada. [2]Department of Cell Biology, New York University School of Medicine, New York, NY, USA. [3]Université Clermont Auvergne, CNRS, Laboratoire Microorganismes: Génome et Environnement, F-63000 Clermont-Ferrand, France. [4]The Donnelly Centre for Cellular and Biomolecular Research, University of Toronto, Toronto, ON, Canada. [5]Department of Microbiology, New York University School of Medicine, New York, NY, USA. ✉e-mail: aaron.reinke@utoronto.ca

microsporidia infections in honeybees[14]. However, recent reports suggest that fumagillin may be ineffective against *N. ceranae* in honeybees and *E. hepatopenaei* in shrimp[15,16]. In addition, fumagillin causes toxicity in humans and its use in beekeeping is banned in some countries[17]. The other most used therapeutic agent against microsporidia is albendazole, which disrupts β-tubulin function. Several microsporidia species, including *E. bieneusi*, have β-tubulin variants associated with albendazole resistance, and as a result, these species are not susceptible to the drug[18]. Several other approaches for the drug treatment of microsporidia infections have been described, including inhibition of chitin synthase, as well as inhibition of spore firing by blocking calcium channels[18,19]. Microsporidia only grow inside of host cells, making screening for inhibitors challenging. Several screens to identify microsporidia inhibitors have been performed, but these have been limited to less than 100 compounds at a time due to a lack of applicable high-throughput screening assays[20,21].

The model organism *Caenorhabditis elegans* is a powerful system to study infectious diseases and is widely amenable to high-throughput drug screens. *C. elegans* is commonly infected in nature by the microsporidian *Nematocida parisii*, which has been used as model to study microsporidia spore exit, host immunity, and proteins used by microsporidia to interact with its host[22–26]. The infection of *C. elegans* begins when *N. parisii* spores are ingested. Spores then germinate in the intestinal lumen, where their unique invasion apparatus known as a polar tube is fired, enabling sporoplasm deposition into intestinal epithelial cells. Intracellularly, the sporoplasm initiates a proliferative process of multiplication by binary or multiple fission, known as merogony, producing meronts in direct contact with the host cytosol[27,28]. Following proliferation, meronts undergo sporogony to form spores which then exit the worm, with over 100,000 spores being produced by each infected animal[29]. Infection of *C. elegans* by *N. parisii* results in impaired growth, reduced progeny production, and lethality[23,27]. The ease of culturing and growing large numbers of animals, along with easily discernible phenotypes, has made *C. elegans* a powerful platform to identify novel anthelmintic, antibiotic, and antifungal agents[30–37].

To discover additional microsporidia inhibitors, we used the *C. elegans* / *N. parisii* model system to develop a high-throughput, liquid-based drug screening assay in which compounds were scored for their ability to prevent infection-induced progeny inhibition. Using this assay, we screened the Spectrum Collection of 2,560 FDA-approved compounds and natural products, and identified 11 chemical inhibitors of microsporidia infection, which we confirmed using a semi-automated method for quantifying *C. elegans* progeny number in liquid culture. We report that five of these compounds, including the known serine protease inhibitor ZPCK and four compounds that share a quinone structure, prevent microsporidia invasion in *C. elegans* by inhibiting spore firing. Additionally, the iron chelator and topoisomerase II inhibitor dexrazoxane prevents infection progression. Together, this work describes methods to screen thousands of putative microsporidia inhibitors and identifies novel microsporidia inhibitors that either block microsporidia invasion or proliferation.

## Results

### High-throughput screen of 2560 spectrum collection compounds reveals 11 microsporidia inhibitors

To identify inhibitors of microsporidia infection, we developed a novel screening assay for chemical inhibitors based on the observation that *C. elegans* progeny production is greatly inhibited when infected with *N. parisii*[23,38]. In our screening assay, L1 animals in 96-well plates were grown in liquid and infected with *N. parisii* spores for six days at 21 °C (Fig. 1A, see methods). In the absence of *N. parisii* spores, *C. elegans* larvae develop into adults and produce progeny. In the presence of spores, the animals are smaller and produce fewer progeny, providing a convenient visual indication of infection. This inhibition of progeny

production is prevented by the known microsporidia inhibitor fumagillin (Fig. 1B). Using this assay, we tested 2560 FDA-approved compounds and natural products from the Spectrum Collection for their ability to prevent infection-induced progeny inhibition. After incubation with compounds for six days, each well was visually assessed for *C. elegans* progeny production. We identified 25 initial compounds that when added to wells, resulted in the production of more progeny than the vehicle, DMSO, controls. Upon retesting, 11 compounds were confirmed to reproducibly restore *C. elegans* progeny production in the presence of spores (Table S1).

To quantify the inhibitory effect of each compound, we developed a semi-automated approach to quantify progeny number in liquid culture, (Fig. 1A, see methods). Animals were grown in wells as described above, stained with the dye rose bengal, and imaged using a flatbed scanner. Images were processed with consistent parameters to highlight stained animals and analyzed with WorMachine to detect and count the number of animals in each well (Fig. S1A–C)[39]. Counts of animals detected using WorMachine correlated well with those counted manually (Fig. S1D). Additionally, there was good correlation (average $R^2$ of 0.71) between technical replicates (Fig. S1E). This approach is similar to a recently published method, but with the advantage of using a relatively cheap flatbed scanner for imaging[40–42]. We observed that each of the 11 compounds was able to significantly increase the number of progeny produced by animals under infection conditions (Fig. 1C). This effect is even more pronounced when considering that for six of the compounds, there was a significant reduction in progeny production in uninfected animals, indicating moderate host toxicity (Fig. 1D).

Our initial screen identified compounds that could rescue the ability of *C. elegans* to produce progeny in the presence of *N. parisii* spores. To determine whether the compounds have a direct effect on microsporidia infection, we performed continuous infection assays. L1 animals were infected continuously with *N. parisii* spores in the presence of compounds for 4 days at 21 °C in 24-well plates in liquid (Fig. 2A). After incubation, animals were fixed and stained with direct yellow 96 (DY96), a fluorescent dye that binds chitin, a crucial component of both the microsporidia spore wall and *C. elegans* embryos (Fig. 2B)[22,43]. First, we observed that every compound significantly increased the proportion of adult animals containing embryos in the presence of *N. parisii* spores (Fig. 2C). These results are consistent with our data from the initial screen showing that all compounds increased progeny production in the presence of spores (Fig. 1C). Second, we determined that the control inhibitor fumagillin and 9 of the 11 compounds inhibited microsporidia infection, as the proportion of animals with newly formed spores was significantly lower upon treatment with these compounds (Fig. 2D).

### *N. parisii* proliferation is inhibited by dexrazoxane

Microsporidia infection can be inhibited either by blocking invasion, or by preventing proliferation after infection is established. To distinguish between these possibilities, we performed pulse-chase infection assays where L1 animals were infected with *N. parisii* spores for 3 h, washed to remove excess spores, and incubated with compounds for 4 days (Fig. 3A). As expected, given its described mode of action as a MetAP2 inhibitor[17], fumagillin restricted *N. parisii* proliferation (Fig. 3B). Surprisingly, of our 11 compounds, only one, dexrazoxane, inhibited *N. parisii* proliferation (Fig. 3E). Both fumagillin and dexrazoxane restored the ability of animals to make embryos when infected, whereas none of the other compounds were able to do so (Fig. 3D). The effect of dexrazoxane, at a concentration of 60 µM, is especially striking with a ~1200-fold reduction in the proportion of animals with newly-formed spores, compared to just ~2-fold for fumagillin, at a concentration of 350 µM (Fig. 3E). At this concentration of dexrazoxane, we observed no negative effect on uninfected animals (Fig. 1D). Together, these results suggest that of the compounds identified from

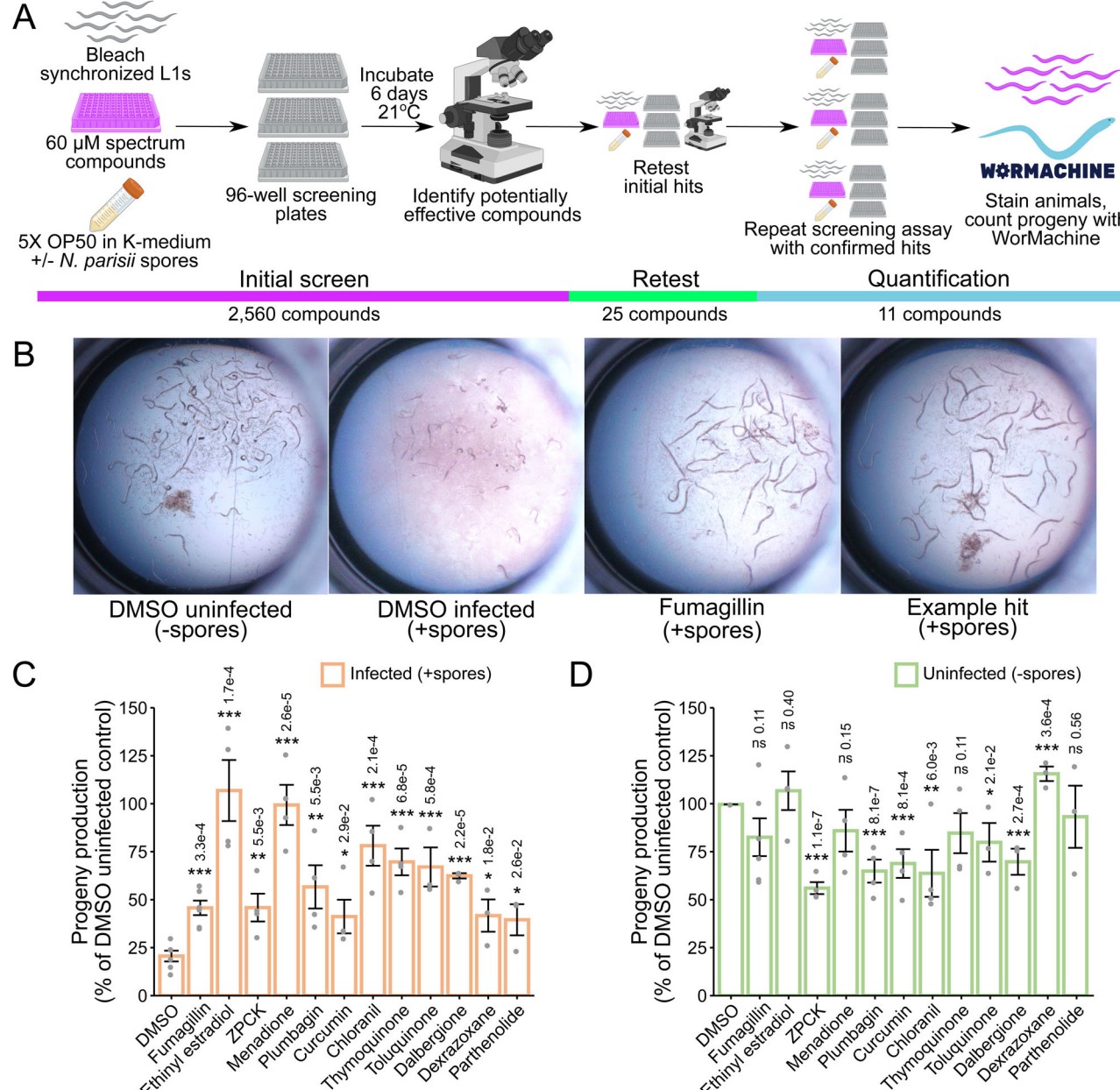

**Fig. 1 | High-throughput drug screen of the Spectrum Collection identifies compounds that restore progeny production in *C. elegans* infected with *N. parisii*. A** Schematic of small molecule inhibitor screen (see methods). Bleach-synchronized L1 animals were incubated with 60 μM of each compound from the Spectrum Collection for 6 days at 21 °C in the presence of *N. parisii* spores. 2,560 total compounds were screened once, yielding 25 initial hits. The initial hits were retested once, yielding 11 confirmed hits. The effectiveness of these 11 compounds was then quantified across multiple replicates of the screening assay using WorMachine. Figure made using Biorender.com. **B** Representative images of wells containing worms. (**B Far Left**) Normal worm growth in the absence of spores. (**B Middle Left**) Microsporidia infection leads to inhibition of progeny production. (**B Middle & Far Right**) Treatment with anti-microsporidial compounds restores progeny production. **C, D** Effect of compounds on progeny production in (**C**) infected and (**D**) uninfected animals (*n* = 6). Progeny levels expressed as a percentage of the DMSO uninfected control. Statistical significance evaluated in relation to DMSO controls using two-sided *t*-tests: ***$p < 0.001$, **$p < 0.01$, *$p < 0.05$, ns = not significant ($p > 0.05$). Bars represent the sample mean, error bars represent $+/-$ 1 standard error from the mean. Source data are provided as a Source Data file.

our screen, only dexrazoxane inhibits *N. parisii* after invasion of *C. elegans* has occurred.

To determine whether dexrazoxane inhibits microsporidia by slowing proliferation or enhancing parasite clearance, we used FISH staining of pulse-chase infected animals with probes specific for *N. parisii* 18S RNA to visualize the sporoplasms and meronts of the earlier stages of infection prior to new spore formation[27]. Although fumagillin and dexrazoxane both resulted in many fewer animals with newly formed spores (Fig. 3E), there was no significant difference in the proportion of animals that had at least some FISH signal (Fig. 3F). When

*C. elegans* is pulse-chase infected with *N. parisii* for 48 h, many large, multinucleated meronts are observed. Worms treated with fumagillin contain both large meronts with many nuclei as well as some parasite cells that fail to progress beyond having 2 nuclei. In contrast, dexrazoxane treatment leads to almost all parasite cells containing 1 or 2 nuclei, and many parasite cells displaying irregular morphology (Fig. 3C). Using images of FISH-stained *N. parisii*, we quantified the area of the infected animal that is covered by sporoplasms and meronts. We observed that fumagillin and dexrazoxane treatment resulted in a significantly reduced pathogen load, with dexrazoxane having the

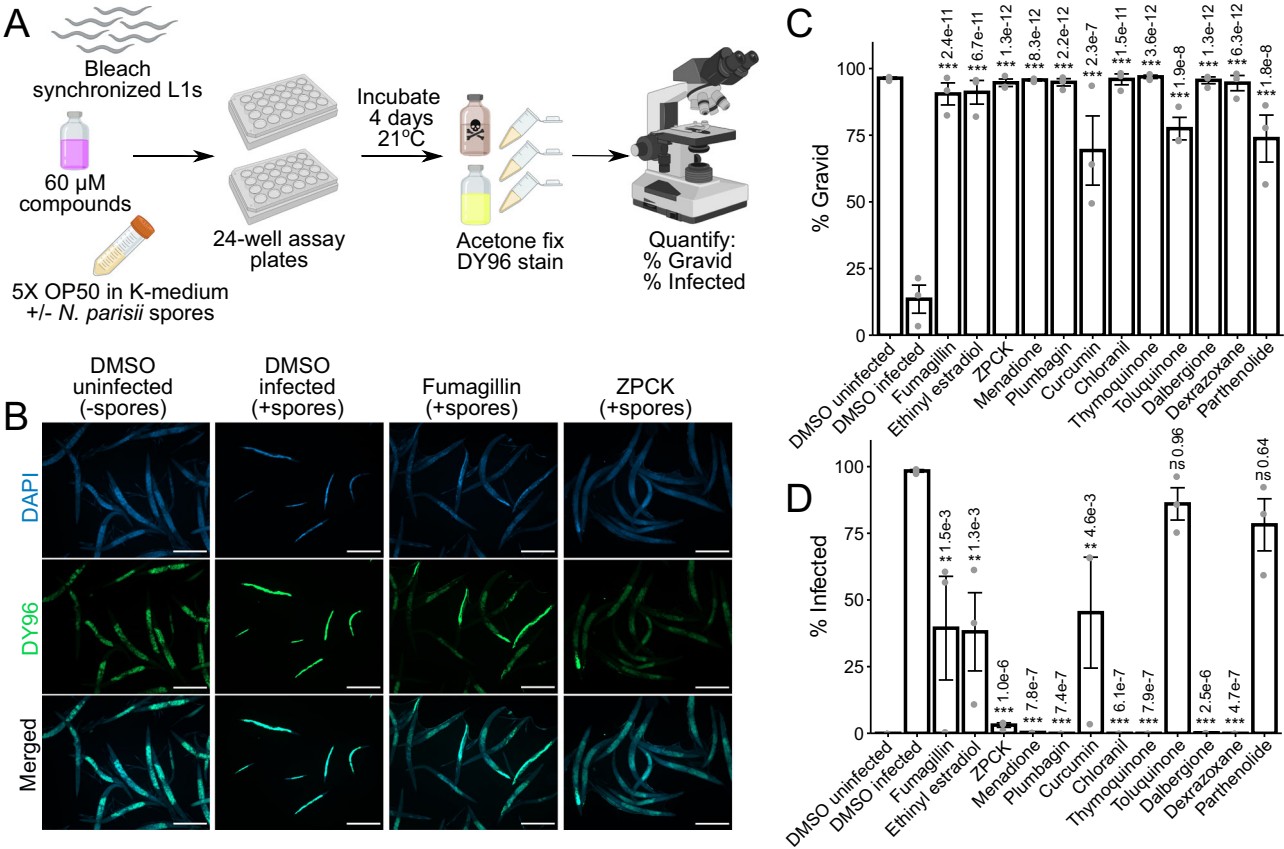

**Fig. 2 | Identified compounds inhibit microsporidia infection. A** Schematic of continuous infection assay (see methods). Bleach-synchronized L1 animals were incubated with compounds for 4 days at 21 °C in the presence of *N. parisii* spores in liquid. Animals were subsequently fixed with acetone, and stained with DAPI (blue) and DY96 (green). Figure made using Biorender.com. **B** Representative images taken at 5x magnification; scale bars are 500 μm. (**B Far Left**) Normal worm growth in the absence of spores. (**B Middle Left**) Microsporidia infection results in production of new spores highlighted in bright green by DY96, and slows growth thereby preventing development of gravid adults. (**B Middle Right & Far Right**)

Treatment with anti-microsporidial compounds reduces formation of new spores, and restores the development of gravid adults. **C** Percentage of animals that contain embryos (*n* = 3, *N* = ≥ 200 animals counted per biological replicate). **D** Percentage of animals that contain newly formed spores (*n* = 3, *N* = ≥ 200 animals counted per biological replicate). Significance evaluated in relation to DMSO infected controls using one-way ANOVA with Dunnett's post-hoc test: ***$p < 0.001$, **$p < 0.01$, *$p < 0.05$, ns = not significant ($p > 0.05$). Bars represent the sample mean, error bars represent + /−1 standard error from the mean. Source data are provided as a Source Data file.

strongest effect (Fig. 3G). Together, these results indicate that dexrazoxane can greatly inhibit *N. parisii* proliferation but does not cause the infection to be cleared.

Dexrazoxane is an iron chelator, and iron levels have been shown to impact the growth of other microsporidia species[44,45]. To determine if iron levels are important for *N. parisii* growth in *C. elegans*, we supplemented our liquid cultures with ferric ammonium citrate (FAC) as a water-soluble iron source[46], and continuously infected L1 animals with a low dose of spores for 4 days. While the proportion of animals with newly formed spores was slightly higher in the supplemented condition, this effect was not statistically significant (Fig. S2A). In contrast, addition of FAC resulted in a small, but significant increase in the proportion of animals containing embryos (Fig. S2B). To determine whether dexrazoxane is likely acting as an iron chelator in our system, we tested another iron chelator, 2,2′-bipyridyl (BP), for anti-microsporidial activity in our continuous infection assay, but did not observe an effect (Fig. 3H and S2C). In addition, FAC supplementation was unable to counteract the anti-microsporidial effect of dexrazoxane (Fig. 3H and S2C). We also tested the *smf-3(ok1035)* mutant strain RB1074 for sensitivity to dexrazoxane. RB1074 has ~50% less iron levels compared to the wild-type N2 strain and displays a striking growth defect upon treatment with the iron chelator BP[46]. However, RB1074 did not display any such growth impairment upon treatment with dexrazoxane (Fig. S3). Taken together, these results demonstrate that

the inhibitory effect of dexrazoxane on *N. parisii* infection is likely independent of its established function as an iron chelator.

## Protease inhibitors and quinone derivatives prevent invasion by inhibiting spore firing

Compounds that displayed strong activity against microsporidia infection in the continuous infection assays, but not the pulse-chase infections, may be preventing invasion by inhibiting spore firing. In *C. elegans*, *N. parisii* spores are ingested, and spore firing is triggered in the intestinal lumen[27,47]. To test if any compounds interfere with this process, we performed spore firing assays on the 7 compounds with the strongest effect in the continuous infection assays (Fig. 2D). L1 stage *C. elegans* were infected with *N. parisii* for 3 h, and then stained with FISH and DY96 (Fig. 4A). This dual staining enables us to determine whether a given spore in the intestine has fired, based on whether it has released its sporoplasm (Fig. 4C). We found that treatment with five compounds (ZPCK, menadione, plumbagin, thymoquinone, and dalbergione) resulted in a significantly reduced proportion of fired spores in the intestine (Fig. 4D). Treatment with these compounds, as well as chloranil, also lead to a notable reduction in the number of sporoplasms per animal (Fig. 4E). The two compounds that inhibit microsporidia proliferation, fumagillin and dexrazoxane, had no significant effect on spore firing, although fumagillin treatment did result in a significant decrease in sporoplasm numbers (Fig. 4D, E).

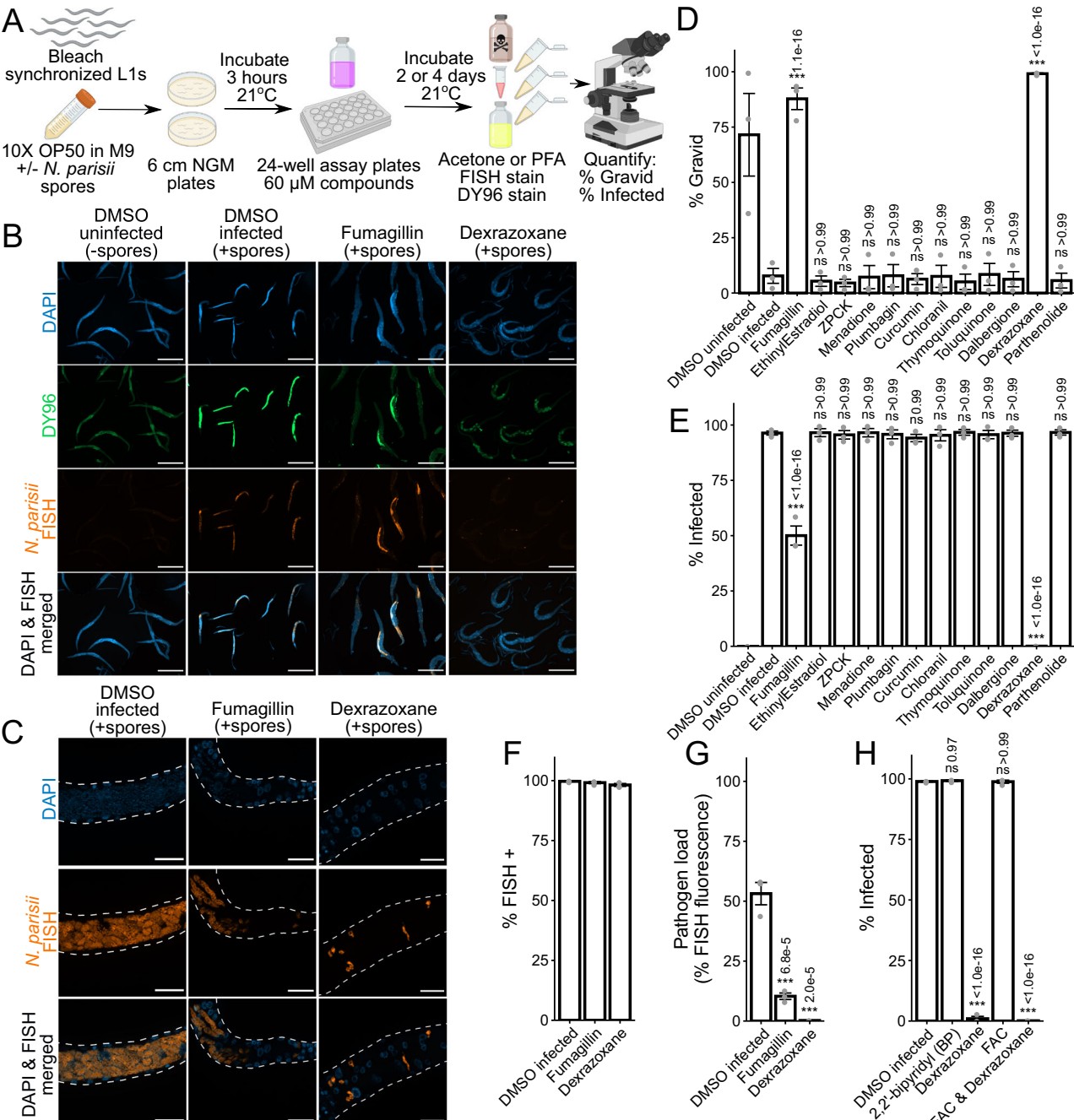

**Fig. 3 | Dexrazoxane prevents microsporidia proliferation. A** Schematic of pulse-chase infection assay (see methods). Bleach-synchronized L1 animals were pulse infected with *N. parisii* spores for 3 h at 21 °C on NGM plates. Excess spores were washed away, and infected animals were incubated with compounds for 2 or 4 days at 21 °C in liquid. Animals were fixed in acetone or PFA, FISH stained, and then stained with DY96 (green) and DAPI (blue). Figure made using Biorender.com. **B** Representative images of acetone-fixed animals 4 days post infection taken at 5x magnification; scale bars are 500 μm. (**B Far Left**) Normal growth in uninfected worms. (**B Middle Left**) Pulse infection results in sporoplasms and meronts highlighted in red by microB FISH probes and new spores highlighted in bright green by DY96. Pulse infection also slows growth thereby preventing development of gravid adults. (**B Middle Right & Far Right**) Treatment of pulse-infected animals with fumagillin or dexrazoxane reduces sporoplasms and meronts, new spores, and restores development of gravid adults. **C** Representative images (z-stack maximum intensity projections; 7 slices, 0.25 μm spacing) of PFA-fixed animals 2 days post infection taken at 63x magnification; scale bars are 20 μm. (**C Left**) In the absence of drug treatment, pulse-infected animals develop large meronts with many nuclei. (**C**

**Middle**) Fumagillin treatment restricts proliferation; both large and small meronts are observed. (**C Right**) Dexrazoxane treatment restricts proliferation; only small meronts with one or two nuclei are observed. **D** Percentage of animals with embryos ($n = 3$, $N = \geq 170$ animals counted per biological replicate). **E** Percentage of animals with newly formed spores ($n = 3$, $N = \geq 170$ animals counted per biological replicate). **F** Percentage of animals with FISH signal ($n = 3$, $N = \geq 170$ animals counted per biological replicate). ANOVA not significant ($p = 0.111$). **G** Quantitation of FISH fluorescence per worm ($n = 3$, $N = 15$ animals quantified per biological replicate). **H** Effects of iron chelation with BP on infection, and effects of iron supplementation with FAC on dexrazoxane activity. Percentage of animals with newly formed spores ($n = 3$, $N = \geq 120$ animals counted per biological replicate) after 4 days of continuous infection is shown. Significance evaluated in relation to DMSO infected controls using one-way ANOVA with Dunnett's post-hoc test: ***$p < 0.001$, **$p < 0.01$, *$p < 0.05$, ns = not significant ($p > 0.05$). Bars represent the sample mean, error bars represent $+/-1$ standard error from the mean. Source data are provided as a Source Data file.

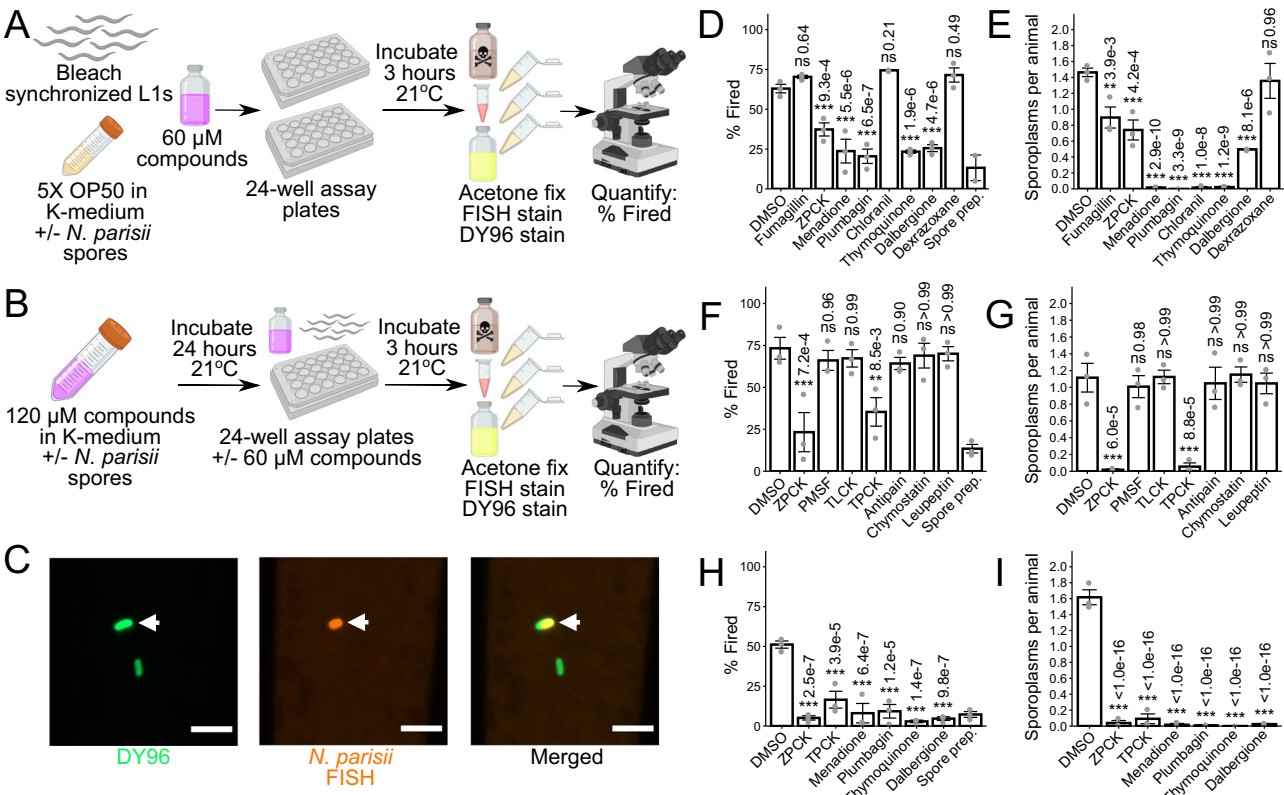

**Fig. 4 | Several compounds prevent microsporidia infection by inhibiting spore firing. A** Schematic of spore firing assay (see methods). Bleach-synchronized L1 animals were incubated with compounds for 3 h at 21 °C in the presence of *N. parisii* spores in liquid. Spore prep control was generated by incubating spores in liquid in the absence of *C. elegans*. Animals and spores were fixed in acetone and stained with microB FISH probe (red) and DY96 (green). Figure made using Biorender.com. **B** Schematic of modified spore firing assay (see methods). Spores were incubated with compounds for 24 h at 21 °C, and then used to infect beach-synchronized L1 animals either with or without prior washing to remove excess compounds. Figure made using Biorender.com. **C** Representative images of unfired and fired spores at 63x magnification; scale bars are 5 µm. Unfired spore is indicated by white arrowhead. **D, E** Effects of microsporidia inhibitors on (**D**) the percentage of fired spores in the *C. elegans* intestinal lumen ($n = 3$, $N = \geq 60$ spores counted per biological

replicate) and (**E**) the average number of sporoplasms per animal ($n = 3$, $N = \geq 40$ animals counted per biological replicate) in a spore firing assay. **F, G** Effects of serine protease inhibitors on (**F**) spore firing and (**G**) sporoplasm invasion in a modified spore firing assay without washing away compounds prior to infection ($n = 3$, $N = \geq 39$ animals and $\geq 60$ spores counted per biological replicate, except for one chymostatin replicate where only 12 animals and 19 spores were counted). **H, I** Effects of spore firing inhibitors on (**H**) spore firing and (**I**) sporoplasm invasion in a modified spore firing assay with compounds washed away prior to infection ($n = 3$, $N = \geq 40$ animals and $\geq 70$ spores counted per biological replicate). Significance evaluated in relation to DMSO controls using one-way ANOVA with Dunnett's post-hoc test: ***$p < 0.001$, **$p < 0.01$, *$p < 0.05$, ns = not significant ($p > 0.05$). Bars represent the sample mean, error bars represent +/−1 standard error from the mean. Source data are provided as a Source Data file.

ZPCK is an irreversible inhibitor of serine proteases, and a microsporidia serine protease has previously been suggested to be involved in spore firing[48,49]. To determine if other serine protease inhibitors can prevent *N. parisii* invasion, we tested three additional small molecules (TPCK, TLCK, and PMSF) and three peptides (Antipain, Chymostatin, and Leupeptin) known to inhibit serine proteases in a modified version of our spore firing assay. For these experiments, we incubated spores with serine protease inhibitors for 24 h to accentuate their effects, and then used the spores to infect L1 stage *C. elegans* for 3 h in the presence of inhibitors (Fig. 4B). Analysis of the proportion of spores fired and the number of sporoplasms per animal revealed that both ZPCK and the structurally related inhibitor TPCK displayed strong inhibition of *N. parisii* spore firing and invasion (Fig. 4F, G).

Inhibitors of spore firing could function either by inhibiting an *N. parisii* spore protein or a *C. elegans* protein necessary for firing and invasion. To distinguish between these possibilities, we incubated spores for 24 h with each of the compounds that inhibited spore firing, washed away excess inhibitor, and then used the treated spores to infect L1 stage *C. elegans* for 3 h in the absence of inhibitors (Fig. 4B). Treatment with all 6 inhibitors (ZPCK, TPCK, menadione, plumbagin, thymoquinone, and dalbergione) significantly inhibited spore firing and sporoplasm invasion (Fig. 4H, I). These results demonstrate that all

of these inhibitors act directly on *N. parisii* spores to prevent firing and invasion, and in the case of ZPCK and TPCK, this likely occurs through the inhibition of a serine protease.

## Identified compounds inhibit multiple microsporidia species
We next determined if the compounds we identified could inhibit infection by other microsporidia species. In addition to being infected by *N. parisii*, *C. elegans* is also infected by *Pancytospora epiphaga*. This species infects the epidermis of *C. elegans* and belongs to the Enterocytozoonida clade which includes the human infecting species *Vittaformae cornea* and *Enterocytozoon bieneusi*[22,50]. To determine if dexrazoxane could inhibit *P. epiphaga* proliferation, we infected *C. elegans* with this species and monitored parasite growth using FISH staining (Fig. 5A). When animals were treated with 350 µM fumagillin or 60 µM dexrazoxane we observed significantly less parasite than in untreated control animals, with the least parasite growth occurring in the animals treated with dexrazoxane (Fig. 5B).

*Anncaliia algerae* infects both mosquitos and humans and belongs to the Neopereziida clade of microsporidia[50,51]. To test whether dexrazoxane could also prevent inhibition of this species, we infected human fibroblast cells with *A. algerae* and visualized infection using FISH (Fig. 5C). We observed a significant dose-dependent

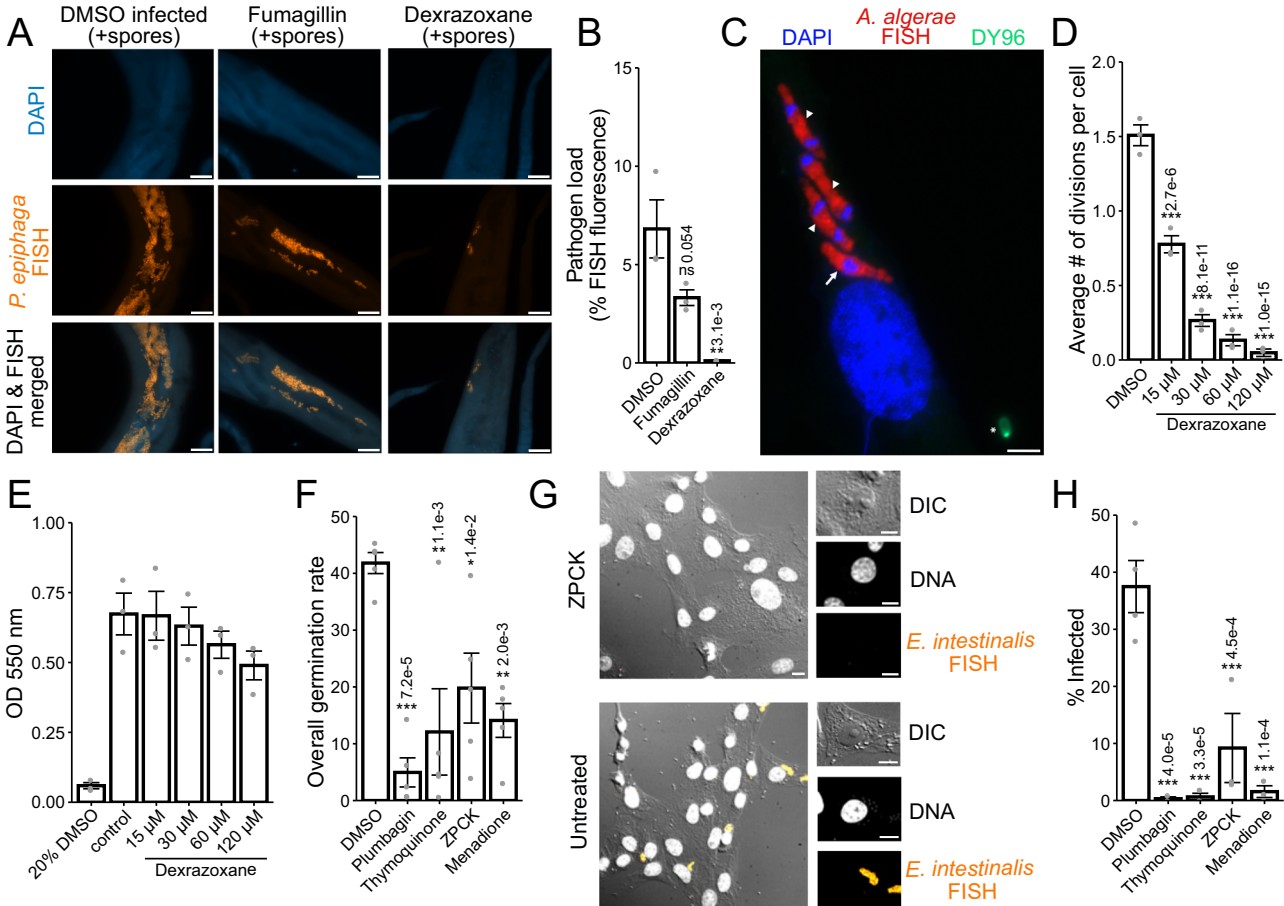

**Fig. 5 | Identified inhibitors display activity against multiple diverse microsporidia species. A, B** Bleach-synchronized L1 animals were pulse infected with *P. epiphaga* spores for 3 h on NGM plates. Excess spores were washed away, and infected animals were incubated with 350 µM fumagillin or 60 µM dexrazoxane for 4 days at 21 °C in liquid. Animals were fixed and stained with DAPI (blue) and a FISH probe (red). **A** Representative images of animals 4 days post infection; scale bars are 25 µm. **B** Quantitation of FISH fluorescence per worm ($n = 3$, $N = 15$ animals quantified in each condition per biological replicate). **C, D** Cells infected with *A. algerae* spores for 3 h with 60 dexrazoxane for 4 days. **C** Representative images of an infected cell stained with dapi (blue), FISH (red), and DY96 (green); scale bars are 5 µm. Meronts undergoing divisions are indicated by arrowheads and meront not currently dividing is indicated by arrow. Spore attached to the cells after initial infection indicated with star. **D** Average number of *A. algerae* divisions per cell ($n = 2$, $N = 30$–243 cells analyzed per biological replicate). **E** Host cell viability

($n = 3$). ANOVA for control and dexrazoxane treatment conditions was not significant ($p = 0.325$) (**F**–**H**) E. intestinalis spores were treated with 60 µM inhibitors for 24 h. Spores were either induced to fire (**F**) or used to infect cells for 24 h and then stained with a FISH probe and DAPI (**G, H**). **F** Percentage of spores that have undergone complete germination ($n = 5$, $N = \geq 100$ spores counted per biological replicate). **G** Representative images of cells infected with *E. intestinalis* that were either untreated or treated with ZPCK; scale bars are 10 µm. DIC, Differential interference contrast microscopy. **H** Percentage of cells infected ($n = 4$, $N = \geq 100$ cells counted per biological replicate). Significance evaluated in relation to DMSO infected controls using one-way ANOVA with Dunnett's post-hoc test: \*\*\*$p < 0.001$, \*\*$p < 0.01$, \*$p < 0.05$, ns = not significant ($p > 0.05$). Bars represent the sample mean, error bars represent + /− 1 standard error from the mean. Source data are provided as a Source Data file.

response of inhibition of parasite division (Fig. 5D). To test whether dexrazoxane was toxic to cells at the concentrations that prevented *A. algerae* proliferation, we monitored cell viability and observed no significant toxicity at the concentrations we tested (Fig. 5E).

*Encephalitozoon intestinalis* infects humans and other mammals and belongs to the Nosematida clade of microsporidia[50,52]. To determine if the compounds we identified could block spore firing in this species, we pre-treated *E. intestinalis* spores with menadione, plumbagin, thymoquinone, or ZPCK, and then carried out an in vitro spore germination assay, in which we can directly observe polar tube firing. We observed a significant decrease in the frequency of germination when spores were pre-treated with each of these compounds (Fig. 5F). To test whether these compounds impact host cell infection, we incubated Vero cells with spores that were either untreated or pre-treated with each compound, and monitored infection with FISH (Fig. 5G). We observed that cells remained largely uninfected when incubated with spores that had been pre-treated with the compounds,

suggesting that all these inhibitors also prevent invasion (Fig. 5H). In the presence of ZPCK, we identified a small number of single invasion events (Fig. 5G), but no replication of parasites. Together our results indicate that a diverse set of microsporidia species can be inhibited by the compounds we identified.

## Discussion
To identify inhibitors of microsporidia, we screened 2,560 compounds for their ability to counteract infection-induced *C. elegans* progeny inhibition. We identified 11 compounds with reproducible inhibitory activity (Table S1). We found that dexrazoxane inhibits *N. parisii* proliferation, whereas the protease inhibitors ZPCK and TLCK, along with 4 quinone derivatives, prevent spore firing and invasion (Fig. 6). Three of the identified inhibitors, curcumin, plumbagin, and thymoquinone, have been previously shown to inhibit microsporidia infection, although in the case of plumbagin and thymoquinone, the compounds were effective after microsporidia had infected cells[20,21]. This contrasts

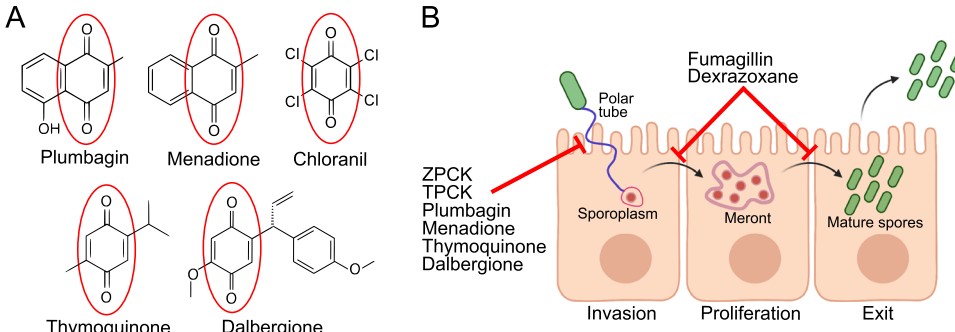

**Fig. 6 | Compound structures and mechanisms. A** Structures of compounds containing a quinone moiety (circled in red) that were identified in the initial screen as inhibitors of microsporidia infection. **B** Microsporidia infection model depicting the stages at which various microsporidia inhibitors act. Two protease inhibitors and four quinone-derivatives were shown to act directly on *N. parisii* spores to prevent spore firing and subsequent invasion of sporoplasms. Fumagillin and dexrazoxane act after invasion has occurred, inhibiting proliferation of sporoplasms and meronts, ultimately reducing parasite burden and preventing the production of microsporidia spores. Figure made using Biorender.com.

with our results, where these compounds only prevent microsporidia invasion. To our knowledge, none of the other compounds we identified have been previously reported to prevent microsporidia infection, demonstrating the value in this unbiased approach to identify novel microsporidia inhibitors.

Dexrazoxane is an FDA-approved drug for the prevention of cardiomyopathies caused by chemotherapeutic drugs in cancer patients[52]. Dexrazoxane can be hydrolyzed into a structure that is similar to the metal ion chelator ethylenediaminetetraacetic acid and is thought to work by reducing the amount of iron complexed with anthracycline chemotherapeutics, thus reducing the number of superoxide radicals formed by such interactions[53,54]. Iron is an important metabolite for microsporidia proliferation. Honey bees infected with *N. ceranae* have reduced levels of iron and the growth of *E. cuniculi* in macrophages was promoted with the addition of iron and inhibited upon addition of an iron chelator[44,45]. Given the established role of dexrazoxane as an iron chelator, it seemed likely that its mechanism as an anti-microsporidial might involve sequestering iron away from sporoplasms and meronts, thus depriving them of a crucial resource required for growth and infection progression. However, our results do not support this. Neither treatment with the iron chelator BP, nor iron supplementation using FAC, had any significant effect on *N. parisii* infection. Additionally, dexrazoxane still exhibited strong anti-microsporidial effects when *C. elegans* were exposed to *N. parisii* in a high iron environment. These results suggest that the anti-microsporidial properties of dexrazoxane are unlikely to be the result of drug-induced alterations in iron homeostasis. This is contrast to lethality of *C. elegans* caused by the pathogenic bacteria *Pseudomonas aeruginosa*, which can be reversed by treatment with the iron chelator ciclopirox olamine, which was included in our screen, but was not observed to have an effect against *N. parisii*[55]. The iron chelator deferoxamine, which inhibits *E. cuniculi* in macrophages, was also included in our screen, but not observed to have activity against *N. parisii*[45]. A second proposed mechanism for dexrazoxane is the inhibition of topoisomerase II[56]. However, several topoisomerase II inhibitors that have previously been used with *C. elegans* (etoposide, teniposide, dactinomycin, and amsacrine), were included in our initial screen, but were not observed to have activity against *N. parisii*[57]. The mechanism of action of dexrazoxane in blocking *N. parisii* proliferation thus remains elusive. We found that dexrazoxane could prevent microsporidia proliferation, at a concentration that has no significant effect on host fitness. However, we do observe a trend towards host toxicity at higher concentrations. Additional analogs of dexrazoxane have been generated and testing dexrazoxane analogs for host toxicity and their ability to inhibit microsporidia could identify more selective compounds[58].

The proteins that control spore firing in microsporidia are unknown[59,60]. Although the in vivo signal for firing is undetermined, several different pH and salt conditions can trigger firing in vitro[59]. Once microsporidia spores are exposed to a host signal, the osmotic pressure of the spore increases forcing the polar tube to be extruded. One protein hypothesized to control spore firing is the subtilisin serine protease which is conserved throughout microsporidia[60]. This protein localizes to the poles of *N. bombycis* spores, and the activated version of the protease localizes to the end of the spore from which the polar tube fires[48]. Here we show that two irreversible serine protease inhibitors (ZPCK and TPCK) can block firing and prevent invasion. Additionally, we identified four quinone derivatives (menadione, plumbagin, thymoquinone and dalbergione) that also block firing and invasion. For all six of these spore firing inhibitors, we observed that firing is still blocked even when the compounds are washed away following their use in the pre-treatment of spores. These results demonstrate that the inhibitors likely act directly on the spores to block firing, even when the trigger for firing occurs inside of the host animal. As spore firing is the initial step in invasion, blocking spore firing is a promising potential strategy to prevent microsporidia infections in agricultural animals.

The *C. elegans* / *N. parisii* system provides a powerful tool to both uncover and characterize novel microsporidia inhibitors. Our approach is high-throughput, relatively cheap, and scalable, allowing the screening of even more diverse collections of small-molecule libraries[61]. Our screen is advantageous as it is performed with wild-type animals and pathogens, and by screening based on host fitness, this approach only identifies compounds with minimal host toxicity[35]. *C. elegans* is also infected by *P. epiphaga*, providing the opportunity to test the specificity of compound inhibition on multiple species of microsporidia in the same host[22]. Several of the compounds we identified also inhibited other microsporidia species, demonstrating the power of using *N. parisii* to discovery novel microsporidia inhibitors. As evidenced by our results, *C. elegans* is a very useful host for determining at what stages of microsporidia infection the identified inhibitors are acting[23]. Several strategies to identify inhibitor targets have been successfully used with other eukaryotic parasites, including generating and sequencing parasite variants that are resistant to inhibition or through thermal proteome profiling[62–64]. *C. elegans* is also likely to be useful for future work to identify inhibitor targets in *N. parisii* using these approaches.

## Methods
### Chemical sources
The Spectrum Collection of FDA-approved compounds and natural products was obtained from MicroSource. For retesting and

post-screen analyses, individual compounds were obtained from Sigma-Aldrich and MicroSource.

### *C. elegans* strains, maintenance, and bleach-synchronization

Animals were cultured as previously described[65]. All experiments were performed with the wild-type N2 strain of *Caenorhabditis elegans* unless otherwise indicated. The *smf-3(ok1035)* mutant strain RB1074 was obtained from the CGC. To generate bleach-synchronized L1 stage animals, L4 animals were first picked onto 10 cm NGM plates and grown for ~96 h at 21 °C. Worms were then washed off plates with M9, and treated with ~4% NaClO / 1 M NaOH solution for 1–3 min to extract embryos, then washed with M9. Embryos were incubated in M9 for 18–24 h at 21 °C to allow hatching.

### *N. parisii* strains and spore preparation

*Nematocida parisii* spores were prepared as previously described[23]. Briefly, large populations of infected C. elegans were disrupted with 1-mm-diameter zirconia beads (BioSpec Products Inc). The resulting lysate was then put through 5-μm filters (Millipore Sigma). Spore preparations were confirmed to not contain any contaminating bacteria and were stored at −80 °C. All experiments were performed with *N. parisii* strain ERTm1.

### 96-well plate-based screen for microsporidia inhibitors

*C. elegans* liquid culture methods and 96-well plate-based screening methods were partially adapted from established protocols[32,66]. *N. parisii* spores were prepared as described above, mixed in K-medium (51 mM NaCl, 32 mM KCl, 3 mM CaCl2, 3 mM MgSO4, 3.25 μM cholesterol) with 5x saturated OP50-1 *E. coli*, and added to each 96-well culture plate. 96-well Spectrum Collection plates were thawed from −80 °C. Using a V&P Scientific 96-well pinning tool, 300 nL of DMSO-dissolved compounds were pinned into screening plate columns 2–11, and 300 nL of DMSO was pinned into screening plate columns 1 and 12 to be used for DMSO infected, DMSO uninfected, and fumagillin controls. Bleach-synchronized L1s were prepared as described above, mixed in K-medium with 5x saturated OP50-1 *E. coli*, and added to the screening plates. The final volume in each well was 50 μL, with ~100 L1s, *N. parisii* spores at a final concentration of 6250 spores/μL, DMSO at a final concentration of < 1%, and compounds at a final concentration of 60 μM except for fumagillin (used at a concentration of 350 μM in all experiments). Screening plates were covered with adhesive porous film, placed in parafilm wrapped humidity boxes, and incubated at 21 °C, shaking at 180 rpm for 6 days. After incubation, progeny production was scored manually by visual inspection of the screening plates. Initial hits were rescreened for reproducibility at a final concentration of 60 μM in the same manner using individual compounds resuspended in DMSO. The inhibitory effect of compounds that passed rescreening were quantified in separate experiments using semi-automated methods described below.

### Semi-automated quantification of progeny production

Bleach-synchronized L1s were treated with compounds +/− *N. parisii* spores in 96-well plates prepared and incubated as described above. Three wells were assayed for each condition for each biological replicate. After incubation, animals were stained by adding 10 μL of 0.3125 mg/mL Rose Bengal solution to each well using an Integra VIAFLO 96 Electronic pipette. Plates were wrapped in parafilm and incubated at 37 °C for 16–24 h. To each well, 240 μL of M9 + 0.1% Tween-20 was added. Plates were then centrifuged for 1 minute at 2200 x g. Next, 200 μL was removed and 150 μL of M9 + 0.1% Tween-20 added to each well. 25 μL from each well was then transferred to a white 96-well plate containing 300 μL M9 per well. Plates were imaged using an Epson Perfection V850 Pro flat-bed scanner with the following parameters (dpi = 4800, colour = 24-bit,.jpg compression = 1). Images were edited in GIMP version 2.10 or later to highlight stained animals

by removing HTML colour codes #000000 and #FFC9AF, applying unsharp masking with the following parameters (radius = 10, effect = 10, threshold = 10), editing hue saturation with the following parameters (For yellow, green, blue and cyan: lightness = 100, saturation = −100. For red and magenta: lightness = −100, saturation = 100), and exporting each well as a single.tiff image with LZW compression[67]. The number of animals in each well was counted using the MATLAB-based phenotypic analysis tool WorMachine with the following parameters (pixel neighbouring threshold = 1, pixel binarization threshold = 30, max object area to remove = 0.003%)[39].

### Continuous infection assays

Continuous infection assays were performed in 24-well assay plates with each well containing a final volume 400 μL, ~800 L1s, 5,000 *N. parisii* spores/μL, 60 μM of each compound except for fumagillin (see above), and DMSO at a final concentration of 1%. Three wells were assayed for each compound for each of three biological replicates. Assay plates were covered with adhesive porous film, placed in parafilm wrapped humidity boxes, incubated at 21 °C, 180 rpm for 4 days, and stored at 4 °C, 20 rpm for 1–2 days. After incubation, samples were acetone-fixed, DY96-stained, and subjected to fluorescence microscopy as described below.

### Pulse infection assays

~8000 bleach-synchronized L1s and 10 million *N. parisii* spores were added to 6 cm NGM plates with 10 μL 10x OP50-1. Plates were dried in a clean cabinet and incubated for a total of 3 hours at room temperature. After pulse infection, animals were washed 1–2x with 1 mL M9 + 0.1% Tween-20 to remove excess spores, then added to transparent 24-well assay plates prepared as described above for the continuous infection assay, except without adding any spores. Three wells were assayed for each compound for each of three biological replicates. Assay plates were treated as described for continuous infection assays, except the incubation period was either 2 or 4 days. Samples incubated for 4 days were acetone-fixed, FISH-stained and DY96-stained, while samples incubated for 2 days were fixed in 4% PFA and FISH-stained. Fluorescence microscopy was performed as described below.

### Spore firing assays

24-well assay plates were prepared exactly as in the continuous infection assay with *N. parisii* spores at a final concentration of 5000 spores/μL, compounds at a final concentration of 60 μM except for fumagillin (see above), and DMSO at a final concentration of 1%. For all spore firing assays containing TPCK, spores at a concentration of 10,000 spores/μL were incubated for 24 h at 21 °C in K-medium with compounds at a concentration of 120 μM, and DMSO at a concentration of 2%. These spores were then used to prepare assay plates exactly as in the continuous infection assay with final concentrations as stated above. For all spore firing assays where excess compound was removed prior to infection, spores were washed 3x with 1 mL K-medium before being used in assay plate preparation. Three wells were assayed for each compound for each of three biological replicates. Assay plates were covered with adhesive porous film, placed in parafilm wrapped humidity boxes, and incubated at 21 °C, shaking at 180 rpm for 3 h. After incubation, samples were acetone-fixed, FISH-stained, DY96-stained, and subjected to fluorescence microscopy as described below.

### Iron chelation and supplementation

24-well assay plates were prepared exactly as in the continuous infection assay with *N. parisii* spores at a final concentration of 5000 spores/μL (normal dose) or 78 spores/μL (low dose), 2,2'-bipyridyl (BP) or dexrazoxane at a final concentration of 60 μM, water-dissolved ferric ammonium citrate (FAC) at a final concentration of 6.6 mg/mL, and DMSO at a final concentration of 1%. Assay plates were

treated as described for continuous infection assays. After incubation, samples were acetone-fixed, DY96-stained, and subjected to fluorescence microscopy as described below.

## RB1074 drug sensitivity tests

~1000 RB1074 or N2 bleach-synchronized L1s were added to 6 cm NGM plates top plated with 120 μL 10x OP50-1, 2,2'-bipyridyl (BP) or dexrazoxane at a final concentration of 60 μM, and DMSO at a final concentration of 1%. Plates were dried in a clean cabinet for ~1 h at room temperature, and then incubated at 21 °C for 3 days. After incubation, live animals were imaged using a Leica Microsystems dissecting scope.

## DY96 staining, fluorescence in situ hybridization (FISH), and microscopy

Post-incubation, samples were washed 1–2x with 1 mL M9 + 0.1% Tween-20 to remove excess OP50, fixed in 700 μL acetone and stored at −20 °C, or fixed in 500 μL PFA solution (4% PFA, 1x PBS, 0.1% Tween-20) and stored at 4 °C until ready for subsequent steps. Samples were then washed 1–2x with 1 mL 1xPBS + 0.1% Tween-20 and 1x with 1 mL hybridization buffer (900 mM NaCl, 20 mM Tris HCl, 0.01% SDS). 100 μL FISH staining solution (5 ng/μL FISH probe in hybridization buffer) was added, and samples were incubated for 18–24 h at 46 °C. The *N. parisii* 18S rRNA-specific microB FISH probe (ctctcggcactccttcctg) conjugated to Cal Fluor Red 610 (LGC Biosearch Technologies) was used[27]. After FISH incubation, samples were washed 1x with 1 mL wash buffer (50 mL hybridization buffer + 5 mM EDTA) to remove excess FISH probe. 500 μL DY96 staining solution (20 μg/μL DY96, 0.1% SDS in 1xPBS + 0.1% Tween-20) was added, and samples were incubated for 1 hour at 21 °C. After DY96 incubation, samples were suspended in EverBrite™ Mounting Medium with DAPI, and subjected to fluorescence microscopy using a ZEISS Axio Imager 2 at 5x–63x magnification with Zeiss Zen 2.3. For continuous infection assays, FISH staining steps were omitted, and just stained with DY96.

## Quantification of FISH fluorescence

FISH fluorescence was quantified using FIJI version 2.1.0[68]. Animal area was determined by outlining boundaries based on DAPI staining. Minimum fluorescence threshold for FISH signal was set to 4000 to exclude auto-fluorescence and background staining of embryos while maximizing inclusion of FISH-stained *N. parisii* sporoplasms and meronts. The percentage area of animal covered by FISH signal was calculated.

## *P. epiphaga* infection assays

Spores of *P. epiphaga* strain JUm1396 were prepared the same as *N. parisii*. For infections, ~8000 bleach-synchronized L1s and 80 million *P. epiphaga* spores were added to 6 cm NGM plates with 4 μL 10X OP50-1. Plates were dried and incubated for a total of 3 h at room temperature. After the pulse infection, the animals were washed 2X with M9 + 0.1% Tween-20 to remove excess spores, then added to transparent 24-well assay plates. Wells contained a total volume of 400 μL of K-medium + 5x OP50-1 mixed with ~800 worms. DMSO was added to wells for a final concentration of 1%, fumagillin was added for a final concentration of 350 μM, and dexrazoxane was added for a final concentration of 60 μM. Assay plates were covered with adhesive porous film, placed in parafilm wrapped humidity boxes, and incubated at 21 °C, 180 rpm for 4 days. Samples were fixed in 4% PFA and FISH-stained with 5 ng/μL of FISH probe specific to *P. epiphaga* 18S RNA (CAL Fluor Red 610-CTCTATACTGTGCGCACGG). Fluorescence microscopy and quantification of FISH fluorescence was performed as described for *N. parisii* infections.

## Determining the effect of dexrazoxane on *A. algerae* cell division

*A. algerae* spores were collected from culture media and washed with sterile water to remove host cells as previously described[69]. Human

Fibroblasts (HFF) cells (ATCC SCRC-1041) were seeded on coverslips at a density of $5 \times 10^4$ cells/well in MEM medium supplemented with 10% inactivated FBS, 2 mM glutamine, 2.5 μg.mL⁻¹ amphotericin B, 100 μg mL⁻¹ streptomycin, 100 U.mL⁻¹ penicillin, 25 μg.mL⁻¹ gentamicin, at 37 °C in a humidified 5% $CO_2$ atmosphere. Once confluence was reached, cells were infected with $1 \times 10^6$ spores of *A. algerae* for 1 h. After 3 washes with culture medium, infected cells were incubated for 30-35 h at 30 °C in culture medium containing either 0, 15, 30, 60 or 120 μM of Dexrazoxane. Two biological replicates were performed, with either 2 or 3 coverslips tested per replicate. After overnight fixation with methanol at −20 °C, parasites were FISH-stained using a Cy3-labeled probe specific to *A. algerae*, following the protocol described[69]. Samples were also stained with DAPI and DY96. Effects of Dexrazoxane on parasite divisions was evaluated by counting the number of meronts in each infected host cell using a ZEISS Axio Imager 2 microscope. The number of divisions for each infected cell was obtained by the following formula: (Number of meronts currently dividing × 2) + (number of meronts not currently dividing −1).

## Determining cytotoxicity of dexrazoxane

HFF cells were seeded in 96-well-microplates at a density of $10^4$ cells/well in the culture medium described above at 37 °C in a humidified 5% $CO_2$ atmosphere for approximately 48 h to reach confluence. The medium was then replaced by fresh culture medium containing different concentrations of Dexrazoxane (15, 30, 60, and 120 μM). Negative control (cells with culture medium only) and positive control (cells with 20% DMSO diluted in medium) were also included. Each condition was tested in 3 separate experiments, in 6 wells per experiment. Dexrazoxane cytotoxicity was evaluated using the tetrazolium dye MTT as previously described[69]. Briefly, the cells were incubated with or without dexrazoxane for 48 h at 37 °C and 5% $CO_2$. 22 μL of the MTT labelling reagent (final concentration 0.5 mg/mL, Sigma-Aldrich) were then added to each well and the plates were incubated for 2 h in a humidified atmosphere (e.g., 37 °C, 5% $CO_2$). After medium removal, the purple formazan crystals were solubilized using 100 μl of the solubilization solution (DMSO:Isopropanol (1:1)) into each well. The absorbance of the formazan product was measured at 550 nm (Multiscan FC, ThermoScientific).

## *E. intestinalis* spore propagation and preparation

*E. intestinalis* (ATCC 50506) were grown in Vero cells (ATCC CCL-81) using Dulbecco's Modified Eagle's Medium with high glucose (DMEM) supplemented with nonessential amino acids (1X) and 10% heat-inactivated fetal bovine serum (FBS) at 37 °C and with 5% $CO_2$. At 70%-80% confluence, parasites were added into a 75 cm² tissue culture flask and the media was switched to DMEM supplemented with 3% FBS. Cells were allowed to grow for ten days and medium was changed every two days. To purify spores, the infected cells were detached from tissue culture flasks using a cell scraper and placed into a 15 ml conical tube, followed by centrifugation at $1300 \times g$ for 10 min at 25 °C. Cells were resuspended in sterile DPBS and mechanically disrupted using a G-27 needle. The released spores were purified using a Percoll gradient. Equal volumes (5 mL) of spore suspension and 100% Percoll were added to a 15 mL conical tube, vortexed and then centrifuged at $1800 \times g$ for 30 min at room temperature. The purified spore pellets were washed three times with sterile DPBS and further purified in a discontinuous Percoll gradient. Briefly, spore pellets were resuspended in 2 mls of sterile DPBS and layered onto a 10 ml four-layered percoll gradient (2.5 mls 100% Percoll, 2.5 mls 75% Percoll, 2.5 mls 50% Percoll, 2.5 mls 25% Percoll), centrifuged at $8600 \times g$ for 30 min at RT. Spores that separated into the fourth layer (100% percoll) were carefully collected and washed twice in 10 mls of sterile DPBS at $3000 \times g$ for 5 min at RT. Purified spore pellets were stored in sterile DPBS at 4 °C for further analyses.

## E. intestinalis germination and infection assays

Purified *E. intestinalis* spores ($2.3 \times 10^7$ spores) were treated with compounds at a final concentration of $60\,\mu M$ or DMSO at a final concentration of 0.6%. For both spore firing and infectivity assays, spores were incubated with compounds for 24 h at room temperature.

For polar tube germination assays, $0.3\,\mu l$ of purified *E. intestinalis* spores was placed on poly-L-lysine treat slides and allowed to air dry briefly. Next, $3\,\mu l$ of pre-warmed germination buffer (140 mM NaCl, 5 mM KCl, 1 mM $CaCl_2$, 1 mM $MgCl_2$, and 5% (v/v) $H_2O_2$ at pH 9.5) was added to the slide and sealed with a #1.5 $18 \times 18$ mm coverslip. PT firing occurs within ~30 s to 1 min of adding germination buffer to the spores. PT firing was imaged on a ZEISS Elyra 7 microscope with a Zeiss C-Apochromat 40x/1.2 water objective with a Dual PCO.Edge 4.2 sCMOS camera. All imaging was performed at 37 C. Z stacks were collected at $0.12\,\mu m$ spacing. Germinated spores were defined as those in which the polar tube was released. At least 100 spores were counted per condition.

For measuring infectivity rates, Vero cells were grown on 12 mm diameter, #1.5 coverslips and infected with spores for 24 h. Cells were fixed in 4% PFA in PBS-T (0.1% Tween 20) for 45 min at room temperature and then processed for FISH. Prior to mounting, cells were stained with NucBlue to visualize host and parasite DNA. Coverslips were mounted onto slides using ProLong Diamond Antifade (ThermoScientific) and sealed. All samples were imaged on a Nikon W1 spinning disc confocal microscope with a Nikon Apo 60×1.40 Oil objective and dual Andor 888 Live EMCCD cameras. Z stacks were collected at $0.3\,\mu m$ spacing. At least 100 cells were counted per condition.

## Statistical analysis

All statistical analyses were conducting using Microsoft Excel 365 and R version 3.6.1 or later accessed via RStudio version 1.2.5019 or later[70–72].

## Reporting summary

Further information on research design is available in the Nature Research Reporting Summary linked to this article.

## Data availability

All data needed to evaluate the conclusions in the paper are present in the paper, Supplementary Materials, and Source Data file. Source data are provided with this paper.

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

## Acknowledgements

We thank Hala Tamim El Jarkass and Alexandra R. Willis for providing helpful comments on the manuscript and Hala Tamim El Jarkass for the illustration in Fig. 6. This work was supported by the Natural Sciences and Engineering Research Council of Canada Undergraduate Student Research Awards (to B.M.), Canadian Institutes of Health Research grant no. 400784 (to A.R.), 173448 (to P.R), and 313296 (to P.R.), Centre for Collaborative Drug Research pilot project fund (to A. R.), Connaught New Researcher Award (to A.R.), an Alfred P. Sloan Research Fellowship FG2019-12040 (to A.R.), a Canada Research Chair award (to P.R.), Pew Biomedical Scholars, PEW-00033055 (to G.B.), Searle Scholars Program, SSP-2018-2737 (to G.B.), NIAID R01AI147131 (to G.B.), NIGMS R35GM128777 (to D.C.E.). Some strains were provided by the CGC, which is funded by NIH Office of Research Infrastructure Programs (P40 OD010440) and we thank WormBase.

## Author contributions

B.M., J.K., A.R., and P.R. designed the inhibitor screen. B.M. performed all the experiments with *N. parisii*. W.Z. and A.R. designed the *P. epiphaga* experiments with W.Z. carrying out the experiments. A.D. and H.E.A. designed and carried out the *A. algerae* experiments. N.A., D.E., and G.B. designed the *E. intestinalis* experiments with N.A. carrying out the experiments. B.M. and A.R. analyzed the data and co-wrote the paper with edits from all of the authors.

## Competing interests

The authors declare no competing interests.
