## [Peer Review File · Nature Communications]

High-throughput small molecule screen identifies inhibitors of microsporidia invasion and proliferation in *C. elegans*Reviewer #1 (Remarks to the Author):

This is a very interesting article that describes a system for screening compounds for activity for microsporidia using a *C. elegans* system. The authors have identified several compounds from a compound library with activity. Of these compounds that have reproducible activity, six inhibit spore germination with 4 having a quinone ring structure and 2 being serine protease inhibitors. In addition one compound (dexrazoxane) was shown to inhibit replication (infection progression). This is significant for the field as the system provides a moderate throughput screen for new drugs and the identified drugs (in particular dexrazoxane) are useful leads for new therapeutic approaches (which are needed as not all microsporidia are sensitive to the currently available therapeutic agents).

The demonstration of the quinone compounds ability to inhibit germination is well thought out and developed. Quinones are known to affect the presence of free radicals, but the mechanism by which these compounds produce this effect on germination was not elucidated by this manuscript.

The 2 serine protease inhibitors (ZPCK and TPCK) may work by inhibiting a subtilisin serine protease that has been shown in *N. bombicis* to localize to the location on the spore coat where germination occurs.

The authors have provided a reasonable exploration of the mechanism of action of dexrazoxane and have demonstrated that iron chelation is not involved in the observed antimicrosporidia effect. This is interesting as the prior mechanism of action of dexrazoxane in its use in cancer therapy toxicity mitigation is via iron chelation. The other activity of dexrazoxane is that of a topoisomerase II inhibitor. They have not done much to investigate this mechanism of action other than noting that the compound library contained several other topoisomerase II inhibitors in the compound library and none of them had activity against microsporidia. This lack of activity could be due to pharmacologic issues (solubility, differences in the microsporidia topoisomerase II binding sites, etc). Dexrazoxane could also have another target. Overall, while the primary purpose of the paper is to describe the screening assay; it would have significantly strengthened the paper if a mechanism of action against microsporidia for this drug was identified.

Minor clarification

1. In the initial screen is semi-automatic analysis used or is the initial scoring done by the observer using a microscope (as suggested by the figure flow diagrams) ? This needs to be more clearly stated in the methods.

Reviewer #2 (Remarks to the Author):

Reinke and colleagues have utilized their experimental system involving microsporidia infection of *C. elegans* to screen for inhibitors of microsporidia invasion and proliferation. A systematic screening strategy, taking advantage of the investigators' familiarity with the details of their microsporidia infection system, identified 11 candidate inhibitors that were characterized further to define a compound, dexrazoxane, which inhibits proliferation to an even greater degree than the antimicrosporidial compound fumagillin, and six compounds that may inhibit spore firing. There are very few compounds with activity against microsporidia—albendazole? Thus, the study provides a potentially important approach to the identification of candidate molecules and may thus be of general interest.

The *C. elegans* work appears solid, and the compounds are of interest. I have one major

comment. Considering the likely toxicity of the compounds—such as quinones-- which the authors see even in the context of *C. elegans*, there is the question of selective toxicity. That is, to be useful potential drugs, the question of whether at the concentrations microsporidia are inhibited, is there generalized toxicity to animal cells? The issue of relative toxicity to *C. elegans* may be relevant if the compounds are to be used to protect *C. elegans* from microsporidia, but presumably, the purpose of the screen is to identify compounds that can be used with different hosts. *C. elegans* is relatively impermeable to chemical compounds, often requiring 1000x greater concentrations due to their cuticle and other barriers to diffusion, and in this case, these properties may protect the host from toxicity. The question that needs to be addressed is whether the compounds exhibit toxicity to mammalian cells in culture or yeast cells at the concentrations used. This should be conducted at the concentrations where anti-microsporidial activity is observed.

Reviewer #3 (Remarks to the Author):

As discussed in the text, the therapy against microsporidiosis is still scarce, especially when considering the control of microsporidia related to agriculture. This work presents a very interesting technology for the screening of substances capable of preventing the infection or multiplication of microsporidia, speeding up the identification of possible therapies for these fungi, a very promising fact. The *C. elegans* model used in this study indirectly evidences the action of substances used to target the microsporidium *N. parisii*, by quantifying the offspring of *C. elegans* and measuring the impact caused by the fungus on its progeny.

The most notable result was the selection ability demonstrated by the methodology, 2,560 compounds were tested for their ability to control microsporidium infection in the *C. elegans* model. Of those, 11 compounds were initially eligible. I suggest that only these compounds are listed in table 1.

Dexrazoxane inhibited the intracellular proliferation of *N. parisii*, while protease inhibitors and quinone derivatives will prevent invasion by inhibiting spore firing. Does the employed methodology aim to identify the fungistatic activity of compounds? Could the authors claim that the drugs have fungistatic activity?

As a suggestion, it would be interesting to compare the data obtained in this model with cell culture using other species of microsporidia.

The methodology is promising for the area, the work brings significant and solid contributions. The results support the conclusions.

Just a few suggestions are posted below:

Line 65 into intestinal epithelial cells (27, 28). Intracellularly, sporoplasms grow into meronts and differentiate into spores

To be more precise regarding the pathogen's biological cycle, my suggestion is:

Intracellularly, the sporoplasm initiates a proliferative process of multiplication by binary or multiple fission, known as merogony, producing merons. Following the proliferative process, merons suffer sporogony and then result in spores

We thank the reviewers for their comments. We have provided a revised manuscript (where all changes are highlighted) to address the reviewers concerns. Because of complications due to COVID, including both restrictions to laboratory space and impact on researchers, this revision has taken longer to complete than anticipated. Points raised by the reviewers have been incorporated into the manuscript (including an additional figure) and we have provided a point-by-point response to each of the reviewer's comments.

REVIEWER COMMENTS

Reviewer #1 (Remarks to the Author):

This is a very interesting article that describes a system for screening compounds for activity for microsporidia using a *C. elegans* system. The authors have identified several compounds from a compound library with activity. Of these compounds that have reproducible activity, six inhibit spore germination with 4 having a quinone ring structure and 2 being serine protease inhibitors. In addition one compound (dexrazoxane) was shown to inhibit replication (infection progression). This is significant for the field as the system provides a moderate throughput screen for new drugs and the identified drugs (in particular dexrazoxane) are useful leads for new therapeutic approaches (which are needed as not all microsporidia are sensitive to the currently available therapeutic agents).

The demonstration of the quinone compounds ability to inhibit germination is well thought out and developed. Quinones are known to affect the presence of free radicals, but the mechanism by which these compounds produce this effect on germination was not elucidated by this manuscript.

The 2 serine protease inhibitors (ZPCK and TPCK) may work by inhibiting a subtilisin serine protease that has been shown in *N. bombycis* to localize to the location on the spore coat where germination occurs.

The authors have provided a reasonable exploration of the mechanism of action of dexrazoxane and have demonstrated that iron chelation is not involved in the observed antimicrosporidia effect. This is interesting as the prior mechanism of action of dexrazoxane in its use in cancer therapy toxicity mitigation is via iron chelation. The other activity of dexrazoxane is that of a topoisomerase II inhibitor. They have not done much to investigate this mechanism of action other than noting that the compound library contained several other topoisomerase II inhibitors in the compound library and none of them had activity against microsporidia. This lack of activity could be due to pharmacologic issues (solubility, differences in the microsporidia topoisomerase II binding sites, etc). Dexrazoxane could also have another target. Overall, while the primary purpose of the paper is to describe the screening assay; it would have significantly strengthened the paper if a mechanism of action against microsporidia for this drug was identified.

We thank the author for their kind comments. As the reviewer suggested, one potential target for dexrazoxane in microsporidia is topoisomerase II. We attempted to determine if the topoisomerase II ortholog in *N. parisii* is relatively more sensitive to dexrazoxane. We tried to do this by complementing a topoisomerase II deletion in budding yeast. This assay has been used previously to show the effect of mutations in the human ortholog of topoisomerase II on drug

resistance. However, the *N. parisii* ortholog could not complement this essential deletion in yeast, so we could not assay the protein for drug sensitivity. Thus, to determine the mechanism of dexrazoxane inhibiting microsporidia, unbiased approaches are likely going to be necessary. Although we feel that these approaches are beyond the scope of this paper, we describe in the discussion in lines 308-312 several different approaches that have worked in other eukaryotic parasites, and are likely to be useful with *N. parisii*.

Minor clarification

1. In the initial screen is semi-automatic analysis used or is the initial scoring done by the observer using a microscope (as suggested by the figure flow diagrams) ? This needs to be more clearly stated in the methods.

The initial screen was done by manual observation. These compounds that passed this primary screen were then retested using semiautomated analysis. This has been clarified in the methods in lines 367-371.

Reviewer #2 (Remarks to the Author):

Reinke and colleagues have utilized their experimental system involving microsporidia infection of *C. elegans* to screen for inhibitors of microsporidia invasion and proliferation. A systematic screening strategy, taking advantage of the investigators' familiarity with the details of their microsporidia infection system, identified 11 candidate inhibitors that were characterized further to define a compound, dexrazoxane, which inhibits proliferation to an even greater degree than the antimicrosporidial compound fumagillin, and six compounds that may inhibit spore firing. There are very few compounds with activity against microsporidia—albendazole? Thus, the study provides a potentially important approach to the identification of candidate molecules and may thus be of general interest.

The *C. elegans* work appears solid, and the compounds are of interest. I have one major comment. Considering the likely toxicity of the compounds—such as quinones-- which the authors see even in the context of *C. elegans*, there is the question of selective toxicity. That is, to be useful potential drugs, the question of whether at the concentrations microsporidia are inhibited, is there generalized toxicity to animal cells? The issue of relative toxicity to *C. elegans* may be relevant if the compounds are to be used to protect *C. elegans* from microsporidia, but presumably, the purpose of the screen is to identify compounds that can be used with different hosts. *C. elegans* is relatively impermeable to chemical compounds, often requiring 1000x greater concentrations due to their cuticle and other barriers to diffusion, and in this case, these properties may protect the host from toxicity. The question that needs to be addressed is whether the compounds exhibit toxicity to mammalian cells in culture or yeast cells at the concentrations used. This should be conducted at the concentrations where anti-microsporidial activity is observed.

We thank the author for their kind comments. To address the reviewer's concerns about host toxicity we tested dexrazoxane in human fibroblasts cells. These experiments showed no significant impact of toxicity up to 120 μ M (Figure 5E). We also infected this cell line with *Anncaliia algerae* and observed that dexrazoxane could inhibit parasite proliferation at concentrations as low as 15 μ M. Together, these experiments demonstrate that dexrazoxane can inhibit microsporidia at concentrations where host cellular toxicity is not observed.

Reviewer #3 (Remarks to the Author):

As discussed in the text, the therapy against microsporidiosis is still scarce, especially when considering the control of microsporidia related to agriculture. This work presents a very interesting technology for the screening of substances capable of preventing the infection or multiplication of microsporidia, speeding up the identification of possible therapies for these fungi, a very promising fact. The *C. elegans* model used in this study indirectly evidences the action of substances used to target the microsporidium *N. parisii*, by quantifying the offspring of *C. elegans* and measuring the impact caused by the fungus on its progeny.

The most notable result was the selection ability demonstrated by the methodology, 2,560 compounds were tested for their ability to control microsporidium infection in the *C. elegans* model. Of those, 11 compounds were initially eligible. I suggest that only these compounds are listed in table 1.

We thank the author for their kind comments. We have modified table S1 so that only the compounds that were quantified are listed.

Dexrazoxane inhibited the intracellular proliferation of *N. parisii*, while protease inhibitors and quinone derivatives will prevent invasion by inhibiting spore firing.

Does the employed methodology aim to identify the fungistatic activity of compounds?

Could the authors claim that the drugs have fungistatic activity?

The inhibitors we identified act through different mechanisms. Compounds that inhibit microsporidia firing prevent the sporoplasms from entering a host cell and are essentially killing the spores, as firing is an irreversible process and sporoplasms are thought to not be viable if outside of a host cell. Conversely, dexrazoxane blocks proliferation, but the same proportion of animals are infected. This experiment suggests that dexrazoxane isn't killing the parasite, but just preventing growth, so we think it is likely that dexrazoxane has fungistatic activity.

As a suggestion, it would be interesting to compare the data obtained in this model with cell culture using other species of microsporidia.

To determine if the compounds we identified have broad inhibitory activity against microsporidia, we tested the compounds we identified against three additional microsporidia species. First, we show that dexrazoxane inhibits *Pancytospora epiphaga* proliferation. This species infects *C. elegans*, but is in the same clade as the human infecting species *Enterocytozoon bieneusi* and *Vittaforma corneae*. Second, as described in our response to reviewer 1, we show that dexrazoxane blocks proliferation of *Anncaliia algerae* in human fibroblasts cells. Third, we show that the spore firing inhibitors we identified (menadione, plumbagin, thymoquinone, and ZPCK) block *Encephalitozoon intestinalis* spore firing and also prevent invasion of Vero cells. Together, our results (Fig. 5) show that the compounds we identified have broad anti-microsporidia activity in different hosts.

The methodology is promising for the area, the work brings significant and solid contributions. The results support the conclusions.

Just a few suggestions are posted below:

Line 65 into intestinal epithelial cells (27, 28). Intracellularly, sporoplasms grow into meronts

and differentiate into spores

To be more precise regarding the pathogen's biological cycle, my suggestion is:

Intracellularly, the sporoplasm initiates a proliferative process of multiplication by binary or multiple fission, known as merogony, producing merons. Following the proliferative process, merons suffer sporogony and then result in spores

This suggested change has now been made.

Reviewer #2 (Remarks to the Author):

The revised manuscript addresses my one criticism of the initial submission, which was that the selectivity of the toxicity of the putative agents was not considered. However, Figure 5E, showing the results of treating human cell lines with compound, does not satisfactorily address the concern. There is substantial loss of viability even with "control" treatment, there appears to be a trend towards toxicity as compound concentration approaches 120uM, and this is within an order-of-magnitude of the proposed active concentration of compound (15 uM). The overall significance and impact of the study is greatly influenced by whether the isolated compounds exhibit selective activity against microsporidia, and thus this should be demonstrated with more compelling data, or the limitations of the compounds in terms of their selectivity should be more clearly described.

Reviewer #3 (Remarks to the Author):

The authors accepted the suggestions and made modifications to improve the presentation. Publication is suggested.

We thank the reviewers for their comments. We have provided a revised manuscript (where all changes are highlighted) to address the reviewers concerns and to incorporate editorial comments. Points raised by the reviewers have been incorporated into the manuscript and we have provided a point-by-point response to each of the reviewer's comments.

REVIEWER COMMENTS

Reviewer #2 (Remarks to the Author):

The revised manuscript addresses my one criticism of the initial submission, which was that the selectivity of the toxicity of the putative agents was not considered. However, Figure 5E, showing the results of treating human cell lines with compound, does not satisfactorily address the concern. There is substantial loss of viability even with "control" treatment, there appears to be a trend towards toxicity as compound concentration approaches 120uM, and this is within an order-of-magnitude of the proposed active concentration of compound (15 uM). The overall significance and impact of the study is greatly influenced by whether the isolated compounds exhibit selective activity against microsporidia, and thus this should be demonstrated with more compelling data, or the limitations of the compounds in terms of their selectivity should be more clearly described.

This figure does not show that there is a loss of viability in the control cells, as the relative viability is compared between the dexrazoxane treatment conditions and the control cells. The data presented in figure 5E mistakenly included the incorrect Y-axis label. The Y-axis label should be "OD 550 nm" and not "Host cell viability". We apologize for this error and have now corrected the labelling in this figure.

We agree with the reviewer that there is a trend towards host toxicity for dexrazoxane, but the ANOVA analysis for this data is not significant ($p= 0.325$). This p-value for this statistical text has been added to the legend for figure 5. We have also added clarification to the discussion (lines 265-268) that states this limitation and mentions that additional dexrazoxane analogs could be tested for host toxicity and their ability to inhibit microsporidia.

Reviewer #3 (Remarks to the Author):

The authors accepted the suggestions and made modifications to improve the presentation. Publication is suggested.

We thank the reviewer for their comments.